



# Inter-relations of precipitation, aerosols, and clouds over Andalusia, Southern Spain revealed by the AGORA Observatory

Wenyue Wang[1,2], Klemens Hocke[1,2], Leonardo Nania[3], Alberto Cazorla[4,5], Gloria Titos[4,5], Renaud Matthey[1,6], Lucas Alados-Arboledas[4,5], Agustín Millares[3,4], and Francisco Navas-Guzmán[4,5]

[1]Institute of Applied Physics, University of Bern, 3012 Bern, Switzerland
[2]Oeschger Centre for Climate Change Research, University of Bern, 3012 Bern, Switzerland
[3]Department of Structural Mechanics and Hydraulic Engineering, University of Granada, 18071 Granada, Spain
[4]Andalusian Institute for Earth System Research (IISTA-CEAMA), 18006 Granada, Spain
[5]Department of Applied Physics, University of Granada, 18071 Granada, Spain
[6]Federal Office of Meteorology and Climatology MeteoSwiss, Payerne, 1530, Switzerland

**Correspondence:** Wenyue Wang (wenyue.wang@unibe.ch)

**Abstract.** The south-central interior of Andalusia experiences intricate precipitation patterns as a result of its semi-arid Mediterranean climate and the impact of Saharan dust and human-made pollutants. The primary aim of this study is to monitor the inter-relations between various factors, such as aerosols, clouds, and meteorological variables, and precipitation systems in Granada using ground-based remote sensing and in situ instruments including microwave radiometer, ceilometer, cloud radar, nephelometer, and weather station. The objective is to identify potential properties of precipitation in the region and in that way improve precipitation forecasting. Over an 11-year period, we detected rain events using a physical retrieval method that employed microwave radiometer measurements. A composite analysis was applied to them to construct a climatology of the temporal evolution of precipitation. It was found that convective rain is the dominant precipitation type in Granada, accounting for 68% of the rain events. The height of the cloud base is mainly distributed at an altitude of 2 to 7 km. Integrated water vapor (IWV) and integrated cloud liquid water (ILW) increase rapidly before the onset of rain. Aerosol scattering at surface level and hence the aerosol concentration is reduced during rain, and the predominant mean size distribution of aerosol particles before, during, and after rain is almost the same. A meteorological environment favorable for virga formation is observed in Granada. The surface weather station detected rainfall later than the microwave radiometer, indicating virga according to ceilometer and cloud radar data. We used rain-day events identified by weather station data to determine precipitation intensity classes and found that light rain is the main precipitation intensity class in Granada, accounting for 72% of the rain-day events. This can be a result of the high tropospheric temperature induced by the Andalusian climate and the reduction of cloud droplet size by the high availability of aerosol particles in the urban atmosphere. This study provides evidence that aerosols, clouds, and meteorological variables have a combined impact on precipitation which can be considered for water resource management and improving rain forecasting accuracy.





## 1 Introduction

Precipitation is vital for human survival and development as it affects the storage and distribution of water resources (Wilheit et al., 1977). Continuous and heavy rainfall is often a trigger for natural disasters such as floods (Hong et al., 2007). Precipitation as a driver of the hydrological cycle has great significance for climate change responses and feedbacks (Kundzewicz, 2008). Aerosols in atmosphere can act as cloud condensation nuclei. The presence of a high concentration of aerosols might increase

cloud cover and decrease the size of cloud droplets (Sarangi et al., 2017; Kant et al., 2021). For heavy precipitation events such as torrential rain, aerosols promote the merging of cloud droplets and the growth of raindrops, increasing the intensity and distribution of precipitation (Hazra et al., 2017). However, for weak precipitation events such as drizzle, aerosols hinder precipitation formation (Alizadeh-Choobari and Gharaylou, 2017). Chemical reactions and turbulence within clouds and the washing effect of precipitation in turn might affect the concentration and size distribution of aerosols (Hobbs, 1993; Chang

et al., 2016; Zheng et al., 2019). The interaction among precipitation, aerosols, and clouds is highly complex that still contains many uncertainties, requiring further research and exploration (Fan et al., 2016).

The inland regions of south-central Andalusia such as Granada, are semi-arid with an average annual precipitation of less than 500 mm and are highly vulnerable to extreme hydrological events due to climate change (AEMET, 2012; Sumner et al., 2003). A decreasing trend in precipitation for the period 1960-2006 has been observed in the Andalusia area of the Spanish

Mediterranean (Ruiz Sinoga et al., 2011). Moreover, this region, like the rest of the Mediterranean region, is also prone to flooding from heavy rains, causing property damage and even casualties (Barriendos et al., 2019; Flores et al., 2022; Belmonte and Beltrán, 2001). On the other hand, the region of Granada experiences numerous Saharan dust outbreaks every year, which have a significant impact on its environment and air quality (Guerrero-Rascado et al., 2009; Navas-Guzmán et al., 2013; Cazorla et al., 2017; López-Cayuela et al., 2023; Fernández et al., 2019). Rosenfeld et al. (2008) indicated that aerosols affect

precipitation and prompt droughts and floods. Understanding the temporal evolution of atmospheric variables during rain events under these conditions is significant for weather forecasting and assessing measures of artificially increased rainfall.

Deep convection contributes to the occurrence of heavy rainfall events (Luu et al., 2022), and aerosols (Saharan dust) can promote the formation of convective clouds and convective rain (Jiang et al., 2018; Gibbons et al., 2018; Khain et al., 2005; Zhao et al., 2022; Xiao et al., 2023; Chen et al., 2020). Jiang et al. (2018) discovered that various types and concentrations of

aerosols have distinct impacts on deep convective clouds. Continentally polluted aerosols tend to enhance convection, while the effect of dust particles varies depending on the region. Employing a spectral bin microphysics model, Gibbons et al. (2018) observed that an increase in Saharan dust particles leads to competition for available water vapour during diffusive growth, resulting in the formation of numerous and smaller crystals and/or droplets. This leads to the release of more latent heat, and promotes convective updrafts and heterogeneous nucleation mechanisms at higher altitudes. As the concentration of

ice nuclei increases, there is a proportional decrease in total surface precipitation. Khain et al. (2005) indicated that aerosols reduce precipitation efficiency of individual cumulus clouds, resulting in the formation of convective clouds and thunderstorms with higher rain rates. Chen et al. (2020) found that aerosols prompt the formation of stronger updrafts to form mesoscale convective systems that enhance vertical mass fluxes and precipitation. As the concentration of aerosols rises, the frequency of



deep convective clouds increases, while the frequency of shallow warm clouds decreases.

The inter-relations between precipitation, aerosols, and clouds are highly dependent on meteorological conditions (Zhu et al., 2023). The overall impact of aerosols on precipitation is contingent upon environmental conditions, such as air humidity and wind shear, which determine whether the increase in aerosols leads to an elevation in condensation production or loss (Khain, 2009). In addition, the formation of virga is influenced by meteorological factors, including cloud height, temperature, humidity, air pressure, wind speed, and air stability. Karle et al. (2023) used ceilometer profiles, soundings, surface rain gauges,

and radar data to identify the seasonal patterns of virga events and assess the influence of surface meteorological measurements. Virga formation is more likely to occur when the cloud base is high, air temperature is elevated, humidity is low, air pressure is low, wind speed is high, and air instability is present, with small raindrops also playing a role (Jullien et al., 2020; Beynon and Hocke, 2022; Airey et al., 2021). Virga is not only linked to severe climatic events like drought, but also to aerosol loads. The average cloud base temperature of virga is below 0°C, which results in effective loss of sublimation and reduced surface

precipitation (Evans et al., 2011). Water vapour that is produced during evaporation or sublimation of virga can be scavenged by aerosol particles (Tost et al., 2006). The virga, which does not reach the ground, is missed in in situ precipitation observations on the surface.

Observational studies of the inter-relations of multiple factors with precipitation systems are challenging due to their complexity and high variability. Aircraft observations have proven effective in studying aerosol and cloud microphysics near cloud bases

and their impact on precipitation (Wehbe et al., 2021), but the technique is expensive, and the number of measurements of this type is very limited, leading to poor representativeness. Ground-based remote sensing and in situ techniques are suitable alternatives with promising results. Various instruments, such as microwave radiometers, ceilometers, Doppler cloud radars, integrating nephelometers, and weather stations, are widely utilized to measure and characterize precipitation, aerosols and clouds with high temporal resolution. Ground-based microwave radiometers have the advantage of being able to measure

vertically integrated atmospheric column (integrated water vapor and integrated liquid water) and rain rate all-day and all-weather (Rose et al., 2005; Wang et al., 2021; Hocke et al., 2019). Vertical profiles from ceilometers and Doppler cloud radars make it possible to study atmospheric dynamics, the formation and evolution of clouds at different altitudes, precipitation types, and the impact of dust aerosols in detail (Airey et al., 2021). Sarna and Russchenberg (2016) showed that the interaction between aerosols and clouds can be efficiently and continuously monitored by leveraging the synergy of lidar, radar, and

radiometers.

The following are the contributions and benefits of this study, which fill the gaps of previous studies.

- Andalusian region with a semi-arid Mediterranean climate as the study area. The region is influenced by the Sahara dust and is highly vulnerable to climate change, resulting in complex precipitation patterns. Nevertheless, there is a shortage of research on the relationship between precipitation and atmospheric variables in regions with similar conditions, such

as the Mediterranean.

- Synergy of cloud radar and other remote sensing instruments. While lidar remote sensing has limitations in the observation of clouds, cloud radar can penetrate clouds to observe the internal structure of clouds and precipitation with





exceptional time and vertical resolution. Cloud radar combined with aerosol lidar can significantly enhance the precision and accuracy of precipitation studies.

– Cloud evolution during rain events. Presenting the evolution and height changes of clouds before, during and after rain can determine the mechanism and type of precipitation.

   – Detection and observation of virga. Virga can cause inaccuracies in weather forecasts in Andalusia.

The goal of this paper is to explore the inter-relations of multiple atmospheric variables with the precipitation system using multisource ground-based observations in Andalusian region. This manuscript is organized as follows. Section 2 describes the

study area, remote sensing and in situ instruments. Section 3 presents the methods used for rain rate retrieval and data analysis. Section 4 discusses the temporal evolution of precipitation, the impact of multiple factors (including aerosols, clouds, water vapor, and meteorological variables) on precipitation systems, as well as the phenomenon of virga rain. Conclusions are given in Section 5.

## 2   Study site description and instrumentation

**2.1   Study area**

Granada is located in Andalusia in southern Spain. It is surrounded by mountain ranges, including Sierra Nevada to the southeast, Sierra de Huétor to the northeast, and Sierra de Almijara to the southwest. Differences exist between the climate inside and around the metropolitan area due to the blocking of mountains (Esteban-Parra et al., 2022). Granada is close to the Mediterranean Sea, and the region exhibits a Mediterranean climate with hot and dry summers. Winters are mild and humid, snow is

rare. The hottest month is July, with an average maximum temperature of 34.2°C, while the coldest month is January, with an average daily temperature of 12.6°C (AEMET, 2012). Rainfall mainly occurs from October to May. Granada is a non-industrial city with less than 230,000 inhabitants and sits in a depression as a plain. The city is affected by local and European man-made pollution, as well as natural dust from the Sahara desert in northern Africa (Valenzuela et al., 2012; Navas-Guzmán et al., 2013; Cazorla et al., 2017). These factors contribute to the complex meteorological characteristics of Granada, particularly

with regards to precipitation.

Measurements presented in this study were performed at University of Granada urban station (UGR), which is part of the Andalusian Global Observatory of the Atmosphere (AGORA, https://atmosphere.ugr.es/en/about/presentation/agora, last access: 13 December 2022). The station is located in the Andalusian Institute of Earth System Research (IISTA-CEAMA) in the southern part of the city of Granada (37.16ºN, 3.61ºW, 680 m a.s.l.). AGORA combines state-of-the-art active and passive

remote sensing and in situ measurements (Benavent-Oltra et al., 2021; Titos et al., 2012). Most of its instruments are part of the Aerosols, Clouds, and Trace gases Research Infrastructure of the European Research Infrastructure Consortium (ACTRIS-ERIC, www.actris.eu, last access: 13 December 2022).



## 2.2 Remote sensing measurements

Tropospheric microwave observations were perfomed using a generation 2 Humidity And Temperature PROfilers (HATPRO-G2) radiometer from Radiometer Physics GmbH (RPG) manfacturer. This instrument performs continuous and automated measurements of the brightness temperature of the sky, with a radiometric resolution of between 0.3 and 0.4 K root mean square error at 1.0 second integration time (Navas-Guzmán et al., 2014). The radiometer employs direct detection receivers in two bands: the water vapor band (K-band) from 22 to 31 GHz and the oxygen band (V-band) from 51 to 58 GHz, with band-
widths between 0.1 and 2 GHz. The half-power beamwidth for the K-band is 3.5° (Rose et al., 2005). A quadratic regression is used to retrieve integrated water vapour (IWV) and integrated cloud liquid water (ILW), etc. Weather sensors on the HATPRO provide some of the inputs to the retrieval process, such as environmental temperature, relative humidity, and pressure (RPG, 2014). The instrument has an additional infrared radiometer for obtaining cloud base brightness temperature. It is part of the MWRnet network (International Network of Ground-based Microwave Radiometers, http://cetemps.aquila.infn.it/mwrnet/, last
access: 13 December 2022) and several studies have shown its capablity to retrieve IWV, temperature profiles, and relative humidity profiles by comparing with radiosondes (Navas-Guzmán et al., 2014; Bedoya-Velásquez et al., 2019; Vaquero-Martínez et al., 2023).

The Jenoptik CHM15k ceilometer measures atmospheric backscatter profiles with a time resolution of 15 seconds since November 2012. It is part of E-PROFILE (European networks of wind and aerosols profiles, https://e-profile.eu/, last ac-
cess: 13 December 2022). The system uses a Nd:YAG narrow-beam microchip laser that operates at 1064 nm and generates pulses of 8.4 uJ at a repetition rate between 5 to 7 kHz. The receiver field of view is 0.45 mrad, and the laser beam has a divergence of less than 0.3 mrad. The system can measure vertically over ranges from 15 to 15000 m with a resolution of 15 m. Full overlap of the laser beam and the telescope is realized at 1500 m above the ceilometer, while 90% overlap can be achieved between 555 and 885 meters a.g.l. using the manufacturer's overlap function calibration. Cazorla et al. (2017) described the
calculation of the range-corrected signal (RCS) for the CHM15k.

The 94-GHz Doppler cloud radar used in this study was manufactured by RPG, based on the frequency-modulation continuous-wave technique. It measures the vertical profile of reflectivity with a time resolution of 3-4 seconds, and ACTRIS-Cloudnet (Illingworth et al. (2007), https://cloudnet.fmi.fi/, last access: 13 December 2022) provides its target classification (Hogan and O'Connor, 2004). The cloud radar operates at a wavelength of 3.19 mm, allowing high-sensitivity detection of clouds and
raindrops. It points to zenith, covering an range between 50 and 12000 m, at a resolution 30 m. It includes accurate absolute calibration and a robust rain protection system. Use of low transmitter power (1.5 W) enables reliable operation and low maintenance. Myagkov and Rose (2016) described the instrument in more detail.

## 2.3 In situ measurements

The UGR weather station measures air temperature, relative humidity, wind speed, pressure, and precipitation with a time
resolution of 1 minute since 2005. A Vaisala HMP60 probe gathers temperature and relative humidity, and a Campbell Scientific model 05103 anemometer monitors wind speed (de Arruda Moreira et al., 2022). The UGR weather station barometer has low



accuracy (1 hPa), so the pressure in this study is only utilized for air density calculations and not for composite analysis. A rain sensor at the weather station can provide precipitation data for 2020 to 2022, with an accuracy of 0.1 mm. Another rain sensor on roof of the building covers the entire analysis period 2010-2022, with an accuracy of 0.2 mm. The two rain sensors show good agreement during the overlap from 2020 to 2022. Air density was derived from weather station data following Wang and Hocke (2022).

Aerosol in situ measurements are registered in AGORA, that additionally to its contribution to ACTRIS also operates in the framework of NOAA/ESRL federated aerosol network (Andrews et al., 2019) and has participated in global analysis of climate-relevant aerosol properties (Laj et al., 2020). The TSI Model 3563 integrating nephelometer measures the light-scattering coefficient of particles at three wavelengths (450, 550, and 700 nm) at dry conditions with a time resolution of 1 minute since January 2006 (Titos et al., 2012; Lyamani et al., 2010). The total wide angular integration is from 7 to 170°, and the backscattering has an angular range of 90 to 170°. A routine maintenance and calibration of the nephelometer is carried out periodically using $CO_2$ and filtered air. Non idealities due to truncation errors and non-lambertian illumination were corrected following the procedure described in Anderson and Ogren (1998).The uncertainty in the scattering coefficient is about 7% (Heintzenberg et al., 2006). A complete description of the procedures applied in data preprocessing and processing is included in Pandolfi et al. (2018).

## 3 Methodology

### 3.1 Rain rate retrieval

The HATPRO software does not cover the retrieval of rain rates, so this study uses an opacity-based physical retrieval method to calculate rain rates. Wang et al. (2021) presented the principle of this retrieval method in detail, and a brief description is given below. The radiative transfer equation for the Rayleigh-Jeans law is:

$$Tb_f = Tb_C \cdot e^{-\tau_f/\mu} + Tm_f \cdot (1 - e^{-\tau_f/\mu}) \tag{1}$$

where $f$ is the microwave channel of HATPRO. The 31 GHz channel is used to estimate rain rates in this study due to its sensitivity to liquid water. $Tb_f$ is the non-rainfall brightness temperature,. $\tau_f$ is the non-rainfall opacity along the zenith. $\mu$ is the cosine of the zenith angle. $Tb_C$ is the cosmic background brightness temperature. $Tm_f$ is the effective mean temperature calculated by the linear equation of temperature, pressure and relative humidity measured by HATPRO's weather sensors (Mätzler and Morland, 2009).

Equation (1) yields the zenith opacity as:

$$\tau_f = -\mu \cdot \ln\left(\frac{Tm_f - Tb_f}{Tm_f - Tb_C}\right) \tag{2}$$

During rain, the zenith opacity $\tau_{rf}$ can be computed iteratively:

$$\tau_{rf}^{(k+1)} = -\mu \cdot \ln\left(\frac{Tm_{rf}(\tau_{rf}^{(k)}) - Tb_{rf}}{Tm_{rf}(\tau_{rf}^{(k)}) - Tb_f}\right) \tag{3}$$



where $\tau_{rf}^{(k+1)}$ is the rain zenith opacity obtained after the $k$th iteration. $Tm_{rf}$ is the rain effective mean temperature. $Tb_{rf}$ is the rain brightness temperature observed by HATPRO. Note that $Tb_f$ cannot be observed during rain. It has to be estimated by temporal interpolation of the opacity obtained during periods of no rain.

The relationship between rain rate $R_f$ and rain zenith opacity $\tau_{rf}$ is expressed as:

$$R_f = \frac{\tau_{rf}}{g_{rf} \cdot Hr} \tag{4}$$

where $g_{rf}$ is the specific and effective rain-absorption coefficient. It is calculated by Mie theory with droplet distributions and parameterized fall velocities. $Hr$ is the vertical distance between the melting layer and the ground, which is calculated by the temperature gradient. Note that determination of rain stop time may be delayed by the outdoor HATPRO due to the water film

on the radome (Wang et al., 2023).

## 3.2 Composite analysis

Composite analysis method (superposed epoch method) is a useful technique for characterizing meteorological or climatic phenomena that are difficult to observe as a whole, such as exploring and understanding the inter-relations between rainfall and atmospheric variables over time (Adams et al., 2013; Zhang et al., 2020; Sapucci et al., 2019; Allan et al., 2020). Composite

analysis consists of two separate data sets of discrete events (e.g. rain events) and continuous time series. A two-dimensional matrix is constructed by intercepting part of the time series that may be affected by the event. The columns of the matrix are time epochs, the rows are events, and the arithmetic mean is computed over the columns. All selected events are expressed as function of their epoch time so that the averaging process yields the mean evolution of a parameter before, during and after rain. This method can highlight the impact of events on various atmospheric variables at critical moments, and weaken the

impact of atmospheric noise (Zheng et al., 2019). The following criteria are used for composite analysis of rain events:

– The timing mark of the onset and duration of rainfall is when ILW (measured by the HATPRO) exceeds 0.6 mm. The threshold for detecting rainfall using ILW typically ranges between 0.1 and 0.6 mm. When raindrops larger than 0.3 mm in diameter are present in the atmosphere, the emissivity increases significantly, making the accuracy of the threshold less critical (Wang and Hocke, 2022). Wang et al. (2023) showed that an ILW of 0.6 mm is suitable for the outdoor

HATPRO G2 to identify rain, which obtains non-rainfall opacities that agree well with an indoor microwave radiometer. In addition, HATPRO in Granada detects rain event start and end times using ILW=0.6 mm (a rain period has ILW >= 0.6 mm), which aligns with ceilometer attenuated backscatter.

– The epoch time 0 represents the onset of rain, and $t$ represents the duration of rain. -1, +1, and $t$+1 represent 1 h before, during, and after rain respectively.

– All compliant rain events were filtered using the HATPRO ILW threshold of no rainfall within 8 hours before the onset of rain and 8 hours after the end of rain for the period October 2010 to November 2021. There are, in total, 694 rain events, including 502 rain events for studying 8 h before and 16 h after rain, 615 rain events for studying 8 h before and 16 h during rain, and 390 rain events (from November 2012 to November 2021) for the composite analysis of ceilometer





data before and after rain. The 8 h before rain can detect the water vapor convection (Adams et al., 2013), and the 16 h
after rain is to have enough time to detect the effect on the atmosphere and ensure a certain number of rain events. Note
that the 16 h during rain (+16) indicates only one rain event included within 16 hours after the onset of rain, and not the
duration of the rain event itself.

## 4 Results and discussion

### 4.1 Inter-relations of precipitation with cloud, water vapour, and aerosols

March 2022 was one of wettest months in 61 years in Spain, with heavy rainfall causing flooding in parts of Andalusian.
Because of this, march 2022 is used here as an example of the potential of the instrumentation to detect and characterize rain
events and their impact on aerosols. Figure 1 provides an example of the vertical distribution of aerosols, clouds, and rainfall,
as obtained from ceilometer and cloud radar measurements over three consecutive days, from March 12th to 14th, 2022. The
ceilometer determines cloud bases and thin clouds as shown in Figure 1a, while the cloud radar can give a full view of clouds
as shown in Figure 1b. Two cumulonimbus clouds with anvil-shaped top can be seen up to 11 km height, which is often accom-
panied by severe weather such as heavy torrential rain and hail. Cloud bases are from 2 to 7 km. The thickness of the clouds
reaches about 9 km, indicating deep convection. The clouds at a height of 3 to 5 km are stratocumulus clouds from about 7:30
UT to 12 UT on March 14, 2022, while the ceilometer signal is mostly fully attenuated above 1 km. Figure 1c shows the cloud
radar target classification from ACTRIS-Cloudnet. It can be seen from Figure 1a that the high attenuated backscatter values
(more than $4 \times 10^{-6} \mathrm{m}^{-1}\mathrm{sr}^{-1}$) from clouds measured by the ceilometer corresponds to rainfall according to Cloudnet target
classification (Figure 1c). The height of the melting layer (0°C level) here is between 2 km and 3 km. A layer of attenuated
backscatter (less than $1 \times 10^{-6} \mathrm{m}^{-1}\mathrm{sr}^{-1}$) from 1 to 2 km before and after rain is due to aerosols. Solar background light affects
the ceilometer signal during daytime (i.e. since 6 to 18 UT).

Due to limited data from the cloud radar, which only covers a small number of rain events since March 2019, this study relies
on the composite analysis of the ceilometer. Figure 2 shows the composite analysis of ceilometer attenuated backscatter before
and after 390 rain events. To show the cloud base distribution more clearly, the height and time resolutions are reduced from
15 m and 15 s to 60 m and 900 s, respectively, by computing the average value. Cloud base height is mainly distributed from 2
to 7 km during the 8 h before rain and around 2 to 3 km during the 6 h after rain. This result closely resembles that of Figure
1. Statistics from ceilometer data reveal that the predominant precipitation type in Granada is convective rain, comprising 68%
of the 390 recorded rain events. The likely reason for this pattern of mostly convective rain in Granada is firstly due to the to-
pography and climate of Granada. Granada is surrounded by mountains, so humid air from the Atlantic via the north-northwest
is forced to rise to higher altitudes to form convection. The high temperature in Granada also contributes to the upward lifting
of the air. Secondly, aerosols have the ability to enhance the creation of convective rain (Jiang et al., 2018). Aerosols increase
the quantity of cloud droplets or ice crystals and slow down their growth rate. This causes the deposition process to release
more latent heat and triggers heterogeneous nucleation mechanisms at higher altitudes, as a result, promotes convection and
has an impact on its development (Gibbons et al., 2018). Figure 2 also indicates that the intensity of the attenuated backscatter,





**Figure 1.** (**a**) Attenuated backscatter measured by a CHM15k ceilometer, (**b**) reflectivity measured by a RPG-FMCW Doppler cloud radar, and (**c**) the radar target classification in Granada, March 12-14, 2022. Height is above sea level.

which is proportional to the aerosol concentration, is lower in the lower troposphere (below 1.8 km a.g.l.) after rain compared to before rain (black box). This is because rainfall is effective in removing aerosols, as explained in more detail below.

Figure 3 shows the composites of IWV and ILW from HATPRO radiometer in Granada. As shown in Figure 3a, IWV remains around 17.5 mm during the 6 to 8 h before rain, but increases rapidly to a maximum value of 23 mm during the 0 to 6 h before rain. Water vapor convection may be responsible for the increase in IWV before rain. Water vapor moving up at low



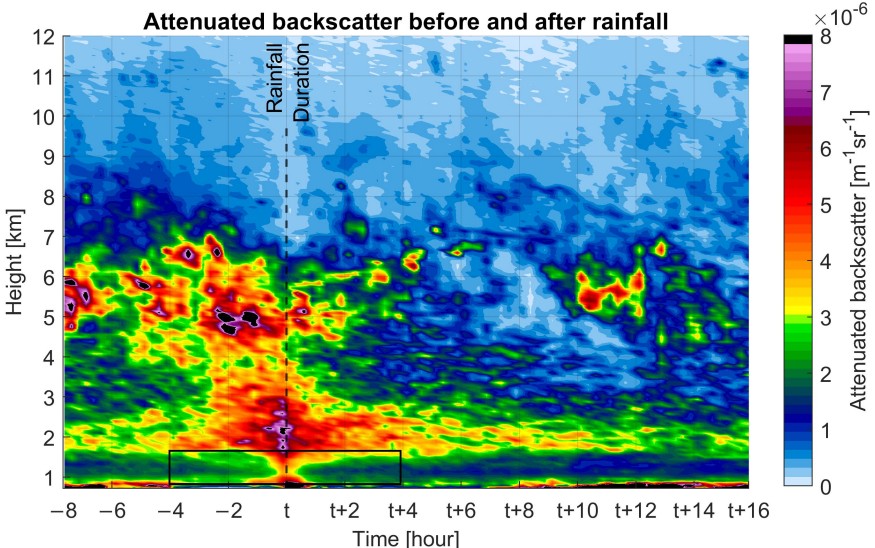

**Figure 2.** Composite of attenuated backscatter measured by a CHM15k ceilometer 8 h before and 16 h after rain. $t$ is the duration of rain. -1 and $t+1$ represent 1 h before and after rain respectively. The range of the black box is below 1.8 km, 4 h before and after rain. Altitude and time resolutions are 60 m and 900 s respectively.

pressure or along a front, collecting around highly hygroscopic condensation nuclei, causing to increased IWV (Koffi et al., 2013). IWV drops sharply and remains at around 16 mm during the 8 to 16 h after rain. The changes of IWV in Granada are slightly different from those observed in Bern, Switzerland (Wang et al., 2021). There is more water vapor in the atmosphere
to form rain in Granada than in Bern. The composite maximum of IWV at the onset of rain in Bern was 21.5 mm, about 1.5 mm lower than in Granada. This provides plenty of vertically rising moist air for convection in Granada. In addition, IWV in Bern is significantly reduced at the end of rain compared to the beginning, which is not the case for Granada. This may be due to the different precipitation types in Granada, whereas Bern has mainly stratiform rain (Wang and Hocke, 2022). Rain events in Bern are more likely to directly remove water vapor from the atmosphere. Figure 3b shows the IWV and the infrared
brightness temperature before and during rain. We can see that IWV remains 28 mm during the 4 h after the onset of rain and then decreases. This may suggest that horizontal transport effects play a major role for water vapor in Granada. It may also be that short-duration rainfall accounts for a relatively large proportion of all rain events, with short-duration rainfall of less than 8 h accounting for 57% in Granada, but only 41% in Bern. The infrared brightness temperature (TIR) reaches its maximum before the onset of rain and then decreases. As shown in Figure 3c, ILW from 0.04 mm to 0.54 mm during the 2 h before rain
and a decrease to 0.03 mm during the 2 h after rain. The sharp increase in ILW is because of the opacity gain as the droplet size (diameter) increases (Wang and Hocke, 2022). The ILW composite peaks at 0.54 mm instead of the ILW threshold of 0.6 mm due to the high temporal resolution of HATPRO. The value is 0.6 mm when calculating the composite median.

Figure 4 shows the composites of scattering coefficient at 550 nm and Ångström exponent (AE) of aerosol particles obtained from the integrating nephelometer in Granada. The scattering coefficient could be used as a proxy of aerosol mass or volume





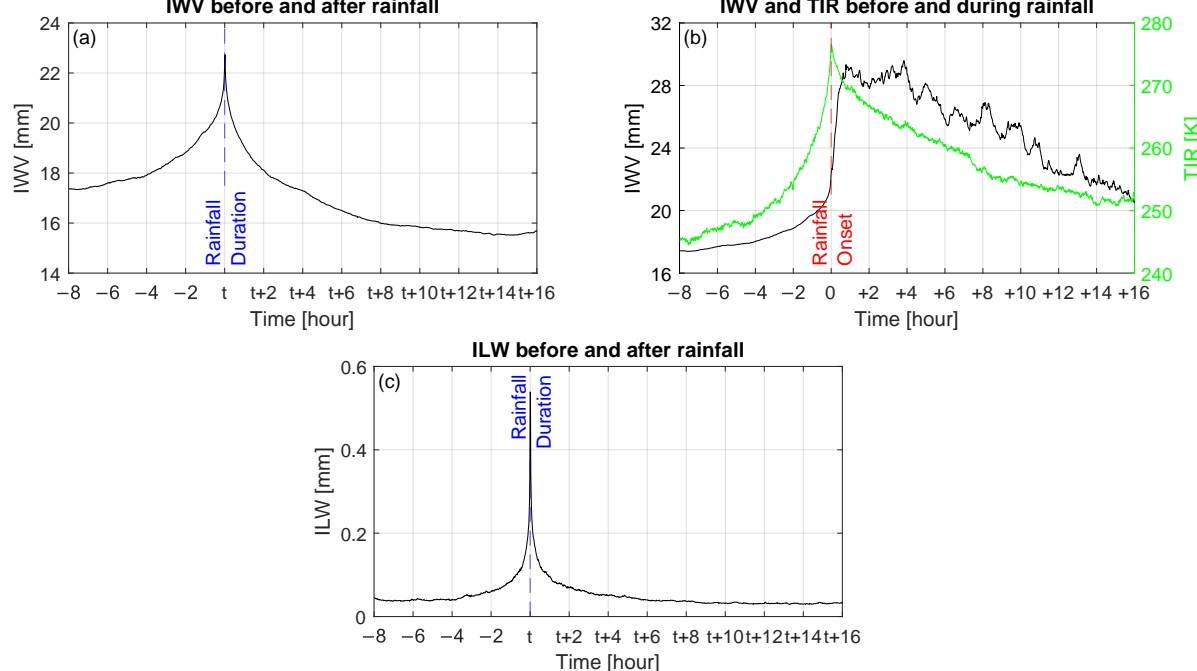

**Figure 3.** Composites of (**a**) IWV 8 h before and 16 h after rain, (**b**) IWV and TIR (green line) 8 h before and 16 h during rain, and (**c**) ILW 8 h before and 16 h after rain provide by HATPRO in Granada.

concentration while the AE provides an estimation of the predominant aerosol mean size at dry conditions. Values of AE>2 indicate a predominance of fine particles while values of AE<1 indicate the predominance of coarse particles. As shown in Figure 4a, the scattering coefficient before rain increases by 2 Mm$^{-1}$ from about 51 Mm$^{-1}$ and then decreases. In Granada, the scattering coefficient at 550 nm peaks between 8:00 and 12:00 UT in the morning as well as between 18:00 and 22:00 UT in the evening due to intense urban emissions (Lyamani et al., 2010). According to statistics, we found that the 4 h before the onset of rain events only accounted for about 28% of the total in these two time periods. Thus, the diurnal variation pattern of aerosols is not the main cause of the significant increase in scattering coefficient 4 hours prior to the onset of rain. The scattering coefficient increases gradually after rain. Due to the removal of aerosols by rain, the scattering coefficient at the end of rain is about 10 Mm$^{-1}$ lower than at the beginning. To the same effect, the scattering coefficient experiences a rapid decrease of 6 Mm$^{-1}$ from approximately 48 Mm$^{-1}$ during the 4-hour rain, followed by a slower decline, and stabilizes at 38 to 42 Mm$^{-1}$ (Figure 4b). As shown in Figure 4c,d, the AE first decreases by about 0.03 from 1.5 before rain, then increases during 4 h rain period, and remains at about 1.51. It increases gradually to 1.58 after rain.

Figure 5 shows a schematic representation of the temporal evolution of aerosol scattering and particle size before, during, and after rain, which makes the variation of scattering coefficient and AE in Figure 4 easy to understand. Before rain, the aerosol concentration first increases and then decreases. This is because the lower relative humidity in the atmosphere before rain increases the surface tension of aerosol particles and makes it easier to suspend in the air. Figure 6 shows the change in relative



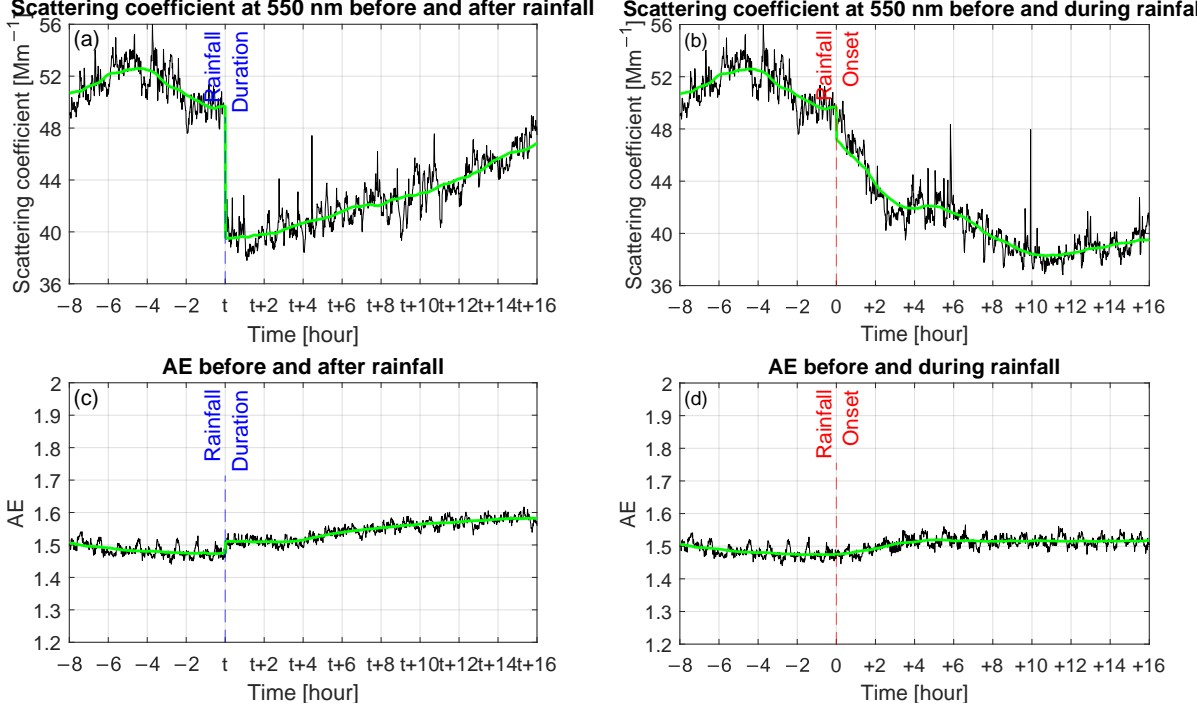

**Figure 4.** Composites of (**a**) scattering coefficient 8 h before and 16 h after rain, (**b**) scattering coefficient 8 h before and 16 h during rain, (**c**) AE 8 h before and 16 h after rain, and (**d**) AE 8 h before and 16 h during rain provide by integrating nephelometer in Granada. $t$ is the duration of rain. 0 is the onset of rain. -1, +1, and $t$+1 represent 1 h before, during, and after rain respectively. The green line is the mean value calculated over a sliding window of 3 h.

humidity. Air movement may also bring aerosols from other regions leading to their increase. Water vapor may condense on certain aerosol particles prior to rainfall, gradually accumulating on their surfaces. After the onset of rain, most aerosol particles are trapped in raindrops and the larger ones settle. Very few larger sized aerosol particles without a water film are also removed from the atmosphere. Smaller particles, due to their low mass, remain suspended in the air and are not easily removed by rain.

As time progresses, larger particles account for a smaller proportion of the remaining particles. After rain, pollution can cause an increase in aerosol concentration. Since AE hardly changes during rain events as shown in Figure 4c,d, the predominant mean size distribution of aerosol particles before, during, and after rain is almost the same. Note that the lack of change in AE from the nephelometer data does not necessarily indicate that the particle size remains constant. In ambient conditions, particles can undergo hygroscopic growth by absorbing water, which would increase their size but may not be detectable by

the instrument. Conversely, the data from the nephelometer indicates that the size of "dry" particles does change, which may suggest that they have the same particle type.



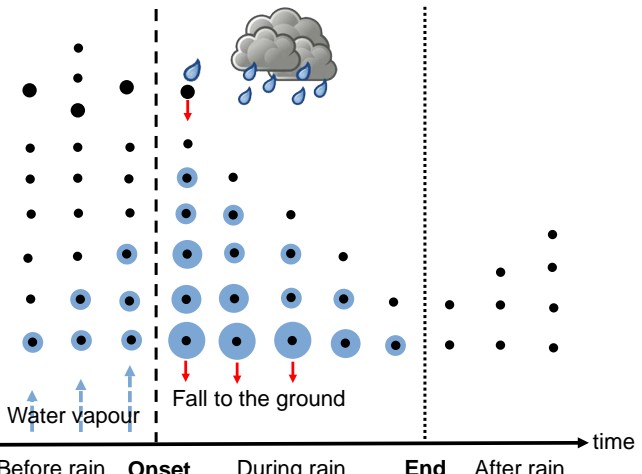

**Figure 5.** Schematic diagram of the temporal evolution of aerosols before, during, and after rain. The dashed line indicates the onset of rainfall, while the dotted line marks its end. The black dots represents aerosol particles, and the blue area represents the water film. The dashed blue arrow represents water vapor, and the red arrow represents particles falling to the ground.

## 4.2 Meteorological effects and virga

Figure 6 shows the composites of surface meteorological variables before and during 615 rain events in Granada. As shown in
Figure 6a, from 4 to 8 h before rain, the ground heating from the sun causes the temperature to rise, air thus warms and expands, reducing the relative humidity. 0 to 4 h before rain, as convective clouds form and rise, the air cools and the surface temperature decreases, increasing relative humidity. Evaporation of surface water increases water vapor content without reaching saturation, which also increases relative humidity. When rain starts, rain droplets falling to the ground in Granada increase the water vapor content, leading to a sudden 1.6% increase in relative humidity and a slight decrease in surface temperature. After the
onset of rain, rain absorbs a large amount of latent heat, resulting in a continuous drop in temperature by 1.3°C during the 12 h of rain, and then an increase in temperature. Rainfall carries away part of the water vapor and slows down the evaporation rate of water, resulting in a decrease of about 2.4% in relative humidity during the 2 h of rain. There is a slight increase in relative humidity over the next 10 h, and then a large decrease in relative humidity. As shown in Figure 6b, before rain, the pressure decreases by 0.9 hPa due to the lifting of air. When it rains, the sudden pressure increase of 0.4 hPa can be due to the
water vapor pressure generated by the evaporation of rain droplets. This value is maintained for a duration of 4 h. From 4 to 9 h after the onset of rain, the pressure continues to rise by 0.3 hPa, and subsequently decreases. Before rain, the wind speed increases by about 0.3 m/s. The pressure gradient between the updraft of convective system and the stronger downdraft around it produces stronger wind speeds. During the 5 h of rain, the wind speed is reduced by the drag force of raindrops on the air, and subsequently remains around 1.4 m/s. As shown in Figure 6c, when rain starts, the rain rate increases rapidly. It peaks at
0.6 mm/s 4 h after the onset of rain, and then slowly decreases to 0.2 mm/s.





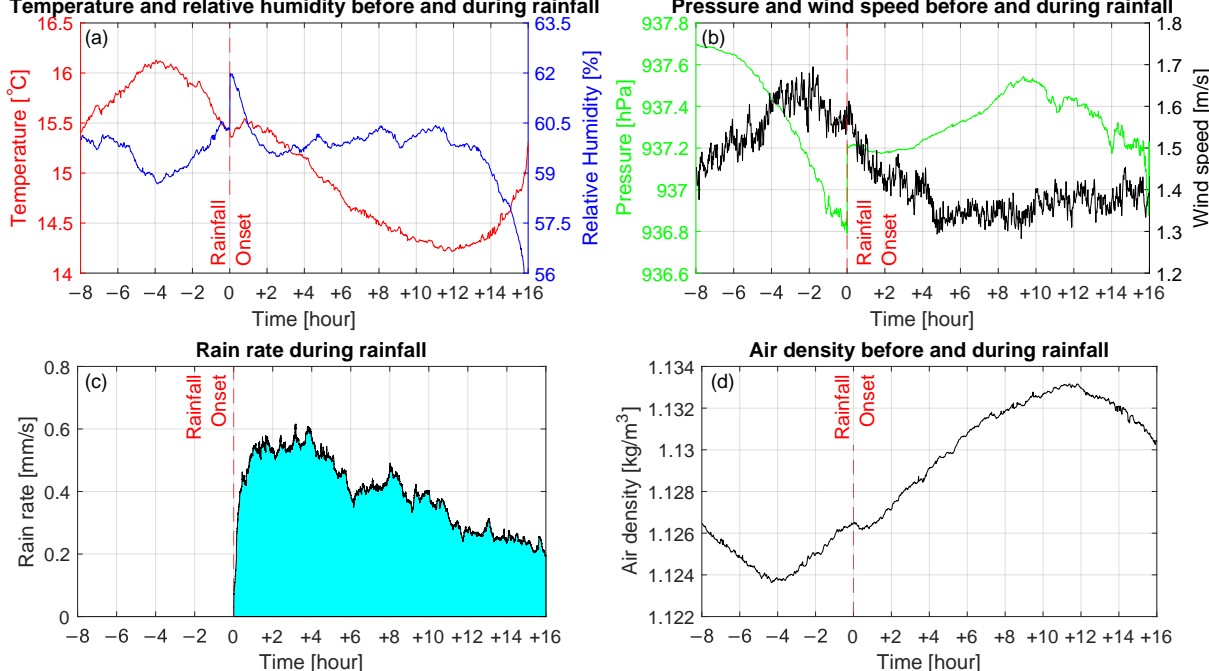

**Figure 6.** Composites of (**a**) temperature (red line) measured by weather station, relative humidity (blue line) measured by weather station, (**b**) pressure (green line) measured by HATPRO weather sensor, wind speed (black line) measured by weather station, and (**c**) rain rate (black line) retrieved from HATPRO 8 h before and 16 h during rain in Granada. 0 is the onset of rain. -1 and +1 represent 1 h before and during rain respectively.

Figure 6d shows composite of air density before and during 615 rain events in Granada. Before rain, air density first decreases and then increases. At 4 h prior to the onset of rain, the air density reaches its minimum value of 1.124 kg/m$^3$. During 12 h after the onset of rain, the air density continues to rise from 1.126 kg/m$^3$ to 1.134 kg/m$^3$, and subsequently decreases. Changes in air density before rain can be one of rainfall precursors and has the potential for nowcasting (Wang and Hocke, 2022). The
more water vapor in the air leads to a decrease in air density due to the lower molecular weight of water vapor compared to the average molecular weight of other gases in the air. Unlike in Bern, the air density in Granada is more dependent on surface temperature (Figure 6a). The changes in these two variables are exactly opposite to each other. Thus, at Granada, the increased surface temperature before rainfall can be taken as a precursor of rainfall too. Overall, this meteorological environment before rain is conducive to the formation of virga.
Although the number of rain events in Granada detected by HATPRO is not small, the amount of rain received by the surface is small. Virga could be the cause. HATPRO, ceilometer, and cloud radar can detect raindrops in the atmosphere, while the rain sensor at the weather station measures the rain falling on the ground. Figure 7 shows ceilometer attenuated backscatter and cloud radar target classification for April 19, 2020 and June 19, 2021, which can reveal cloud and rain information. The time of rain events onset is marked with HATPRO and weather station. As shown in Figure 7a,b, stratocumulus clouds between





2 km to 5 km produce stratiform rain. The onset of the rain detected by HATPRO is 07:24 UT. This time is consistent with the cloud radar target classification. The surface weather station measured the start of rain at 17:12 UT. There is a 10-hour time discrepancy between these instruments in detecting the onset of rain, which could be attributed to the rapid absorption of moisture by the dry air in Granada. Raindrops falling through the dry air may evaporate before reaching the ground, leading to the formation of virga. Unlike cloud radar, almost no rain can be seen from ceilometer attenuated backscatter at 7:00 UT

and 9:00 UT. Cloud radar is more sensitive in detecting rain as it can detect smaller raindrops that gradually evaporate and lose water. These smaller raindrops may not be accurately captured by the ceilometer. As shown in Figure 7c,d, cumulonimbus clouds between 3 km to 11 km produce convective rain. The onset of the rain detected by HATPRO is 02:40 UT. This time is consistent with the ceilometer attenuated backscatter and cloud radar target classification. The weather station measured that it began to rain at 17:15 UT, and the rain lasted only 10 minutes on the ground.

Figure 7e,f shows ceilometer attenuated backscatter and cloud radar reflectivity factor on May 31, 2020. Cumulonimbus clouds between 3 km to 11.5 km produce convective rain. The onset of the rain detected by HATPRO is 13:47 UT. This time is consistent with the ceilometer attenuated backscatter and cloud radar target classification. The surface weather station measured the start of rain at 16:29 UT. As shown in Figure 7e, the virga only lasted a brief period during both 14:00 UT and 15:00 UT, while the virga starting around 15:30 UT continued for approximately 1 h. As shown in Figure 7f, some places in the cloud

at about 8 km have larger values of reflectivity factors.Water droplets or ice crystals from virga can be lifted by convective updrafts into cumulonimbus clouds and continuously grow and coalesce to form rain, such as from 16:29 UT to 17:29 UT. For this rain event, the cumulative rain is not large, only 1.1 mm. This could be due to the decay of deep convective clouds, with weakened updrafts and decreased cloud height. Another possibility is that the concentration of water vapor around the cloud base is higher due to virga, and the warmer temperature and higher cloud height promote continuous collision and merging of

raindrops, resulting in drizzle. Rain-day events are commonly used to evaluate precipitation intensity classes, which represent the total cumulative amount of rainfall in a day (Wang et al., 2021). The weather station provided 889 rain-day events due to its ability to measure the amount of rainfall reaching the ground. Statistics indicate that light rain is the main precipitation intensity class in Granada (accumulation of daily rain less than 5mm), accounting for 72% of these recorded rain-day events. Aerosols may be partly responsible for it. Abundant aerosol particles in Granada acted as cloud condensation nuclei, increas-

ing the number of cloud droplets. This intensified the competition among cloud droplets for water vapor molecules, leading to smaller cloud droplet sizes. Consequently, rain is finer and more disperse, becoming drizzle or light rain.

## 5   Conclusions

In this study, Andalusia, a semi-arid Mediterranean climate region in southern Spain, is chosen as the study area due to its

persistent exposure to Sahara dust and vulnerability to climate change. The precipitation patterns in the region are intricate, as evidenced by instances of severe rainfall causing flooding, as well as inaccurate weather forecasts of precipitation over land. To comprehensively understand the precipitation system in Andalusia, we utilize a combination of ground-based remote sensing





**Figure 7.** Ceilometer attenuated backscatter and cloud radar target classification for (**a** and **b**) April 19, 2020 and (**c** and **d**) June 19, 2021. (**e**) Ceilometer attenuated backscatter and (**f**) cloud radar reflectivity factor on May 31, 2020. Dashed and dash-dotted lines are the onset of rain identified by HATPRO and the weather station respectively.

and in situ instruments to investigate the inter-relations of precipitation on the evolution of various atmospheric variables. The instruments used in this investigation include the microwave radiometer, ceilometer, cloud radar, integrating nephelometer, and



weather station. Rain events and rain-day events were detected by the microwave radiometer HATPRO and weather station for 11 years from October 2010 to November 2021, respectively. The time series of variables obtained from all these instruments is subjected to composite analysis.

First, it was found that convective rain is the main type of precipitation in Granada. Convective rainfall accounts for 68% of the rain events. The vertical distribution of cloud, rain, and aerosol observed by the ceilometer is well interpreted with the

assistance of cloud radar. The cloud base height is primarily distributed in the 2-7 km range 8 h before rain and around 2-3 km during the 6 h after rain. Integrated water vapor (IWV) provide by the microwave radiometer increases rapidly from 0 to 6 h before rain and remains above 28 mm within 4 h after the onset of rain. IWV is not significantly reduced at the end of rain compared to the beginning.

Observations from both the ceilometer and the integrating nephelometer show that aerosols are removed after rain, but the

latter instrument gives more detail. The integrating nephelometer data shows that aerosol scattering increases slightly before rain, followed by a decline with the removal by rainfall, with gradual recovery after rain. There is no significant variation in the predominant mean size distribution of aerosol particles before, during, and after rain.

Before rain, the temperature shows an initial increase followed by a decrease, the relative humidity decreases and then increases, the pressure decreases by 0.9 hPa, and the wind speed increases by 0.3 m/s. At the onset of rain, the raindrops fall onto the

warmer surface and evaporate causing a sudden increase in relative humidity of 1.6% and a pressure increase of 0.4 hPa. The meteorological environment before rain is conducive to the formation of virga. Virga is identified by measuring the time delay between rainfall in the atmosphere and its arrival at the surface, using the microwave radiometer and weather station. The vertical distribution of virga is well observed and showed by the ceilometer and cloud radar. Furthermore, light rain is the main precipitation intensity class in Granada. Light rain accounts for 72% of the rain-day events.

The results of this work on cloud heights before and after rain lead to a better understanding of the formation and evolution of clouds and precipitation in Andalusia. The changes in aerosol scattering and particle size before, during, and after rain reveal the mechanism of aerosol removal and the interaction between aerosol particles and water droplets. Observations and detections of virga can provide knowledge which is required for improvement of local precipitation forecasts, which is crucial for managing droughts, floods, soil erosion, and water resources. Future research could explore the impact of topographic

settings and the proximity of Mediterranean sea on meteorological changes and spatial patterns assessment.

*Author contributions.* WW, KH and FN conceptualized the study. WW performed methods, formal analysis, and software, as well as prepared the original draft. LN, AC, GT, LA, AM, and FN provided support with respect to data curation. RM contributed to the interpretation of the ceilometer results. All co-authors contributed to the interpretation of the results and participated in the manuscript editing and discussion.

*Competing interests.* The contact author has declared that none of the authors has any competing interests.



*Acknowledgements.* The authors thank Christian Mätzler for the software and algorithm support that enabled HATPRO to perform the rain rate retrieval. The first author thanks Alistair Bell for discussions related to the present work.

*Financial support.* The work of Wenyue Wang was supported in part by the China Scholarship Council (CSC) and in part by the Aerosol, Clouds and Trace Gases Research Infrastructure (ACTRIS). Francisco Navas-Guzmán received funding from the Ramón y Cajal program (ref. RYC2019-027519-I) of the Spanish Ministry of Science and Innovation. This work was supported by Grant PID2021- 128008OB-I00

funded by MCIN/AEI/10.13039/501100011033/ FEDER "A way of making Europe", and the project AEROMOST (ProExcel_00204) by the Junta de Andalucía. This work was also supported by the European Union's Horizon 2020 research and innovation program through project ACTRIS.IMP (grant agreement No 871115) and ATMO_ACCESS (grant agreement No 101008004), by the Spanish Ministry of Science and Innovation through projects ELPIS (PID2020-120015RB-I00), NUCLEUS (PID2021-128757OB-I00), and ACTRIS-España (RED2022-134824-E), by the Junta de Andalucía Excellence project ADAPNE (P20-00136), AEROPRE (P-18-RT-3820).This research

was partially supported by University of Granada Plan Propio through Singular Laboratory AGORA (LS2022-1) and Scientific Units of Excellence Program (grant no. UCE-PP2017-02).



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
