# Peer review of "Inter-relations of precipitation, aerosols, and clouds over Andalusia, Southern Spain revealed by the AGORA Observatory"

_EGUsphere, 2023_

## Author Comment (AC1)

**Response to Anonymous Referee #1 (RC1):**

**General comments:**

The subject of the paper is certainly of scientifc relevance, as made clear in the Introduction. I could imagine that the data set the authors present is very valuable and bears potenital for an in-depth analysis of the precipitation processes over Andalusia and their relation of aerosols.

This manuscript does not meet required scientific standards. Essential ones are missing, such as formulation of a clear research question/hypothesis, presentation of comprehensible methods, discussion of the uncertainties of the applied methods, scientifcally sound relation of the results to the derived conclusions. Also, the drawn conclusions as presented here do not really present novel findings as far as I see. Additionally, a number of formulations are scientically unprecise.

**Response 6:**

We sincerely appreciate your dedicated review of our paper and the feedback you have provided. We acknowledge your recognition of the scientific relevance of our paper's topic, and your appreciation of the potential value of the dataset we have presented.

We understand and respect your concerns regarding the scientific standards of our paper. We wish to address these concerns by providing further clarification:

Regarding the research question, our study is centered on addressing the persistent uncertainties surrounding the intricate interactions among precipitation, aerosols, and clouds in the context of Andalusia's semi-arid Mediterranean climate, which is notably influenced by Saharan dust. This question is explicitly stated and forms the foundation of our investigation (please see line 20-41).

For the physical algorithm used in rainfall retrieval for the microwave radiometer HATPRO, we have provided detailed descriptions in our prior publications [1-2]. Your suggestion of scientific imprecision in some formulas is noted, and we kindly request that you specify them, which would aid our discussions and future improvements. The composite analysis method, also better known as the Superposition Epoch Method (SEA), is a well-established technique in atmospheric research [3-5]. SEA simplifies complex meteorological and climatic inquiries by superimposing and contrasting data from different time periods. This method enables the analysis of interactions and trends in atmospheric phenomena. In response to your query about the SEA method's accuracy, we have included standard errors ($\sigma/\sqrt{n}$) for IWV and ILW in the revised manuscript (please refer to Figure 3).

Additionally, we highly prioritize establishing a scientific and logically sound connection between our results and the derived conclusions. Throughout the manuscript, we carefully elucidate this connection using physical principles. For instance, we explain in detail the types of rainfall in Granada and the variations in meteorological parameters before, during, and after rainfall, as well as the causes and progression of the virga phenomenon.

We would like to emphasize that this paper represents a significant scientific innovation and offers novel

findings. To the best of our knowledge, there exist no prior studies that explore the complex aerosol-influenced rainfall characteristics and temporal evolution of rainfall in the Mediterranean Sea using multiple remote sensing and in-situ techniques, and beyond this studies about virga phenomenon are rare.

[1] Wang, W.; Hocke, K.; Mätzler, C. Physical Retrieval of Rain Rate from Ground-Based Microwave Radiometry. *Remote Sens.* 2021, *13*, 2217. https://doi.org/10.3390/rs13112217

[2] Wang, W., Murk, A., Sauvageat, E., Fan, W., Dätwyler, C., Hervo, M., ... & Hocke, K. (2023). An Indoor Microwave Radiometer for Measurement of Tropospheric Water. IEEE transactions on geoscience and remote sensing, 61, 1-13.

[3] Jia, J., Kero, A., Kalakoski, N., Szeląg, M. E., and Verronen, P. T.: Is there a direct solar proton impact on lower-stratospheric ozone?, Atmos. Chem. Phys., 20, 14969–14982, https://doi.org/10.5194/acp-20-14969-2020, 2020.

[4] Rong, P., von Savigny, C., Zhang, C., Hoffmann, C. G., and Schwartz, M. J.: Response of middle atmospheric temperature to the 27 d solar cycle: an analysis of 13 years of microwave limb sounder data, Atmos. Chem. Phys., 20, 1737–1755, https://doi.org/10.5194/acp-20-1737-2020, 2020.

[5] Friederich, F., Sinnhuber, M., Funke, B., von Clarmann, T., and Orphal, J.: Local impact of solar variation on NO2 in the lower mesosphere and upper stratosphere from 2007 to 2012, Atmos. Chem. Phys., 14, 4055–4064, https://doi.org/10.5194/acp-14-4055-2014, 2014.

---

## Author Comment (AC2)

**Response to Anonymous Referee #2 (RC2):**

Thank you very much for your careful review, critical comments and improvements! Please find below the original comments and the authors' response (in blue). Revised sentences are in italics.

The manuscript explores the precipitation properties in Granada using a set of remote sensing and in situ instruments. The study focuses on the relationship between precipitation and aerosols, measured over 11 years. There are minor corrections to be made by the authors, but the overall manuscript is very good and well-written. Thus, I suggest the manuscript to be published.

**Specific comments:**

**Point 1:** Line 5: I would say that the objective is to characterize the precipitation events regarding the aerosol and meteorological properties, with much less emphasis on forecasting. In particular, the forecast aspect of the paper is not discussed enough to have it as a clear aim of the study.

**Response 1:** Thank you! We have removed this sentence.

**Point 2:** Line 15: How many rain-day events were identified by the weather station data? Please make it clear here.

**Response 2:** We have added the number of rain-day events identified by weather stations (please see line 13).

**Point 3:** Line 126: I suggest replacing "etc" with "among other products".

**Response 3:** We have replaced the "etc" (please see line 125).

**Point 4:** Line 135: replace "uJ" by "μJ"

**Response 4:** We have changed the unit (please see line 135).

**Point 5:** I suggest the authors indicate in the manuscript methodology section, together with the start period of measurement, the amount (percentage) of data used in the study for each instrument. It is unclear if there are gaps in the measurements that could add bias to the data.

**Response 5:** Thank you for your suggesttion. The instruments obtain corresponding time series based on rain events, and the missing of radiometer HATPRO data directly affects the number of rain event detections. We added the percentage of missing HATPRO data (please see line 215).

*"22% of data are missing during this time interval due to reasons such as unexpected shutdown of HATPRO."*

We also added the standard errors ($\sigma/\sqrt{n}$, shaded areas) of IWV, ILW, and IR to characterize the deviation of the composite (please see Figure 3).

[Figure]

[Figure]

[Figure]

**Point 6:** Regarding the nephelometer measurements, did the authors correct the data for standard temperature and pressure measurements? What were the humidity conditions in which the data were collected? This information must be explicit in the manuscript.

**Response 6:** Yes, the nephelometer uses pressure and temperature to correct for scattering from aerosol particles (please see line 164-166).

*"Temperature and pressure are measured to calculate the scattering of air molecules. This value is subtracted from the total scattering to isolate the scattering attributed to aerosol particles."*

Data were collected under dry conditions with relative humidity less than 40%. We added this information (please see line 159):

*"The TSI Model 3563 integrating nephelometer measures the light-scattering coefficient of particles at three wavelengths (450, 550, and 700 nm) at dry conditions with relative humidity less than 40%."*

**Point 7:** Line 174: remove the extra ".". After "zenith", add a ";" and remove the ".".

**Response 7:** We have removed the dots and added the comma (please see line 178).

**Point 8:** Lines 269-271: I suggest moving the sentences "The scattering coefficient could be used as a proxy of aerosol mass or volume concentration while the AE provides an estimation of the predominant aerosol mean size at dry conditions. Values of AE>2 indicate a predominance of fine particles while values of AE<1 indicate the predominance of coarse particles." to the in situ measurements section.

**Response 8:** We have moved the sentences to the in situ measurements section (please see lines 160-163).

**Point 9:** Lines 275 - 276: the authors stated: "Thus, the diurnal variation pattern of aerosols is not the main cause of the significant increase in scattering coefficient 4 hours prior to the onset of rain". I suggest the authors add a figure (which could be, for instance, in the supplementary material) confirming this affirmation.

**Response 9:** It is a good idea to add a figure. We plotted the distribution of the number of rain events

(please see Figure 5).

[Figure]

**Figure 5.** Stairs plot of the distribution of the number of rain events (694 in total). The grey areas represent the peak time period of the scattering coefficient at 550 nm between 8:00 to 12:00 UT and 18:00 to 22:00 UT.

We also restate these sentences (please see lines 275-280).
*"In Granada, the scattering coefficient at 550 nm peaks between 8:00 and 12:00 UT in the morning as well as between 18:00 and 22:00 UT in the evening due to intense urban emissions (Lyamani et al., 2010). As shown in Figure 5, rain events mainly start to occur from 12:00 to 18:00 UT in the afternoon. Because of insolation, there is convection of moist air during daytime. In the afternoon hours, the moist air has reached a high altitude, so that formation of cloud droplets and raindrops occurs in the adiabatically cooled air parcel. It also illustrates that the diurnal variation pattern of aerosols is not the main cause of the significant increase in scattering coefficient 4 hours prior to the onset of rain."*